# LIMPACAT: Multi-omics attention transformer for immune prediction in liver cancer using whole-slide imaging

Yen-Jung Chiu [ID]*

Department of Biomedical Engineering, Chang Gung University, Taoyuan, Taiwan

* d000020163@cgu.edu.tw

## Abstract

Characterizing the tumor immune microenvironment from histopathological images offers opportunities for ex vivo immune profiling and prognostic assessment. However, the TCGA-LIHC dataset lacks direct immune cell composition data. Therefore, this study aims to introduce Liver Immune Microenvironment Prediction and Classification Attention Transformer (LIMPACAT), a deep learning framework that leverages whole-slide images (WSIs) to predict immune cell levels relevant to hepatocellular carcinoma (HCC) prognosis. Immune cell compositions were inferred using a deconvolution approach, with bulk RNA-seq profiles simulated from liver-specific single-cell RNA sequencing data and processed with multiple normalization methods. These inferred compositions served as supervision signals to train a multiple instance learning model with an attention transformer. LIMPACAT exhibited ~80% accuracy in classifying immune cell levels from HCC WSIs, showing strong concordance between model prediction and deconvolution-derived estimates. These findings suggest that WSIs can serve as a proxy for immune profiling, facilitating pathology-based tumor microenvironment assessment and supporting personalized therapeutic strategies.

## Introduction

The tumor microenvironment (TME) is a complex system comprising cancer cells and diverse immune cell populations. Interactions between these cells have been shown to influence tumor progression and metastasis [1]. In hepatocellular carcinoma (HCC), the most common primary liver cancer, immune cell composition and function are particularly critical [2]. For instance, elevated levels of memory CD8 + T cells are associated with improved patient outcomes [3]. However, interactions with cancer cells can drive CD8 + T cells into an "exhausted" state, a condition correlated with poorer prognosis in patients exhibiting high levels of exhausted CD8 + T cells [4]. Differences in immune cell subtypes, characterized by distinct surface proteins, potentially exert opposite effects on cancer cells. For example, chimeric antigen

**Data availability statement:** The complete LIMPACAT source code is publicly available from the GitHub repository (https://github.com/holiday01/LIMPACAT.git) and has been permanently archived on Zenodo (https://doi.org/10.5281/zenodo.17168482). TCGA-LIHC whole-slide images are publicly available from the Genomic Data Commons (GDC) Data Portal (https://portal.gdc.cancer.gov/). Single-cell RNA-seq data are publicly available from the Gene Expression Omnibus (GEO) repository under accession GSE189903 (https://www.ncbi.nlm.nih.gov/geo/query/acc.cgi?acc=GSE189903). PBMC bulk RNA-seq validation data are publicly available from the GEO repository under accession GSE107011 (https://www.ncbi.nlm.nih.gov/geo/query/acc.cgi?acc=GSE107011).

**Funding:** This research was supported by grants from the National Science and Technology Council (NSTC), Taiwan (NSTC 112-2222-E-130-003- and NSTC 113-2221-E-130-005-MY3), which assisted in the acquisition of computational facilities essential for this study.

receptor (CAR) T cell therapy involves isolating and activating patient-derived T cells to target cancer cells; however, their function remains constrained by the immuno-suppressive TME. Regulatory T cells (Tregs) suppress antitumor immunity, diminishing CAR T cell efficacy [5]. Similarly, immune checkpoint inhibitors (ICIs) target inhibitory proteins such as PD-1, PD-L1, and CTLA-4, that drive T cell exhaustion, thereby restoring immune function and enhancing antitumor activity [6]. This evidence highlights the importance of characterizing the immune TME to optimize cancer treatment and prognosis. Although flow cytometry can determine immune cell types, it requires large numbers of viable cells and has limited applicability for cancer tissue analysis [7]. To address these challenges, cell composition deconvolution (CCD) algorithms leverage computational biology to enable data reuse and large-scale data analyses of TME influences on cancer [8–11]. Rather than replacing experimental assays, CCD algorithms extract key biological features from existing datasets, thereby supporting the discovery of potential insights into cancer biology. Early CCD models relied on regression model frameworks such as quanTIseq and CIBERSORTx, which depend on predefined immune signatures and therefore struggle to capture nonlinear gene–cell relationships and tumor-specific immune heterogeneity. More recent studies have demonstrated that incorporating single-cell RNA sequencing (scRNA-seq) references with deep learning (DL) architectures substantially improves deconvolution accuracy. In our previous work, we introduced a DL-based CCD model that achieved higher concordance with true immune cell fractions than both CIBERSORTx and quanTIseq [8].

Recent studies show that the availability of scRNA-seq data has increased significantly through public repositories such as NCBI GEO and 10x Genomics. Unlike bulk RNA-seq or microarray platforms, scRNA-seq provides higher precision by enabling cell-level resolution [12,13], allowing the identification of immune cell subtypes and their interactions within tumor samples, such as exhausted CD8＋T cells and FCGR3A+ macrophages in liver cancer [14]. However, scRNA-seq datasets are prone to batch effects, making data integration across diverse sources challenging without appropriate normalization. To mitigate batch effects, several normalization methods have been developed, such as log normalization [15], canonical correlation analysis (CCA) [16], and SCTtransform [15], enabling more accurate and consistent cross-sample comparisons. Effective normalization is crucial for ensuring the reliability of downstream analyses, particularly in studies investigating complex interactions within the tumor microenvironment [17,18].

Recent advances in attention-based DL models have driven significant progress in computational pathology, particularly for whole-slide image (WSI) analysis. The Clustering-Constrained Attention Multiple Instance Learning (CLAM) model leverages attention mechanisms to identify diagnostically relevant regions in WSIs under weak supervision [19]. By assigning differential attention weights, CLAM improves interpretability and data efficiency, which is particularly beneficial in settings with limited annotations [20]. However, CLAM is potentially constrained in capturing finer interregional relationships, limiting its applicability beyond classification tasks [19]. Attention-based convolutional neural networks have also shown promise in meningioma classification

by prioritizing tumor regions identified as critical by pathologists, thereby enabling accurate molecular classification [21,22]. Although beneficial for advancing precision medicine workflows, this approach requires extensive management of multi-layered attention and may be challenging to generalize across tumor types [21]. Meanwhile, the MHAttnSurv model employs multi-head attention to capture diverse morphological patterns in tumor slides, enhancing the robustness of survival prediction. Despite its effectiveness, MHAttnSurv demands high computational resources, limiting its scalability for large-scale WSI applications [23].

Digital histopathology is a critical tool in cancer diagnosis, offering a more accessible and cost-effective alternative to molecular assays [24]. Hematoxylin and eosin (H&E) staining is the most widely applied method, enhancing tissue contrast by differentially staining cytoplasm [25], cell nuclei, extracellular matrix, and other cellular structures [24, 26]. These histopathological findings can provide prognostic insights; however, manual interpretation of large-scale images is impractical. Variations in immune cell abundance further influence patient outcomes, highlighting the need for accurate classification of immune cell levels using digital pathology. Therefore, this aims to develop Liver Immune Microenvironment Prediction and Classification Attention Transformer (LIMPACAT), for predicting immune cell abundance associated with hepatocellular carcinoma (HCC) prognosis directly from pathological images. Given the inherent heterogeneity of the tumor microenvironment, CCD was integrated to generate pseudo-labels from scRNA-seq data, providing reliable immune groupings for weakly supervised training. LIMPACAT could offer deeper insights into immune responses within the tumor microenvironment, enhancing prognostic assessments and expanding the potential of digital pathology.

## Materials and methods

### Study design

This study aims to predict prognosis in patients with HCC using WSIs by characterizing immune infiltration levels. However, direct immune composition labels were unavailable for WSI data. Therefore, scRNA-seq and a deep learning–based cell composition deconvolution (DL-CCD) model were employed to estimate immune cell fractions from bulk RNA-seq data. These estimates were subsequently used to generate pseudo-labels to supervise WSI-based model training (Fig 1).

Training phase: To fully leverage high-resolution scRNA-seq data, we first evaluated normalization strategies to ensure robust cross-dataset integration. For the DL-CCD model, an ensemble of three neural network architectures was implemented, trained and validated on simulated bulk RNA-seq derived from HCC scRNA-seq, and externally tested on real PBMC bulk RNA-seq to confirm accuracy. DL-CCD predictions on TCGA-LIHC bulk RNA-seq were then used to identify immune subsets significantly related to survival outcomes. These survival-associated immune subsets served as pseudo-labels for WSI training. Within the LIMPACAT framework, we benchmarked four multiple instance learning (MIL)-based classifiers to learn immune-related features from WSIs for patient prognosis stratification.

Inference phase: During inference, the trained LIMPACAT model required only WSIs as input. By integrating outputs from multiple MIL classifiers, the framework estimates immune infiltration levels and indirectly stratifies patients into favorable or poor-prognosis groups.

### Data download and preprocessing

ScRNA-seq data were obtained from the GSE189903 dataset, which initially comprised 34 primary liver cancer samples, including HCC and intrahepatic cholangiocarcinoma specimens collected from distinct tumor regions. For this study, only the 20 samples annotated as HCC were analyzed, corresponding to patients represented in the TCGA-LIHC cohort (The Cancer Genome Atlas–Liver Hepatocellular Carcinoma). This restriction ensured disease specificity and minimized confounding from distinct liver cancer subtypes. All scRNA-seq preprocessing and downstream analyses were performed using the Seurat v5 R package [27].

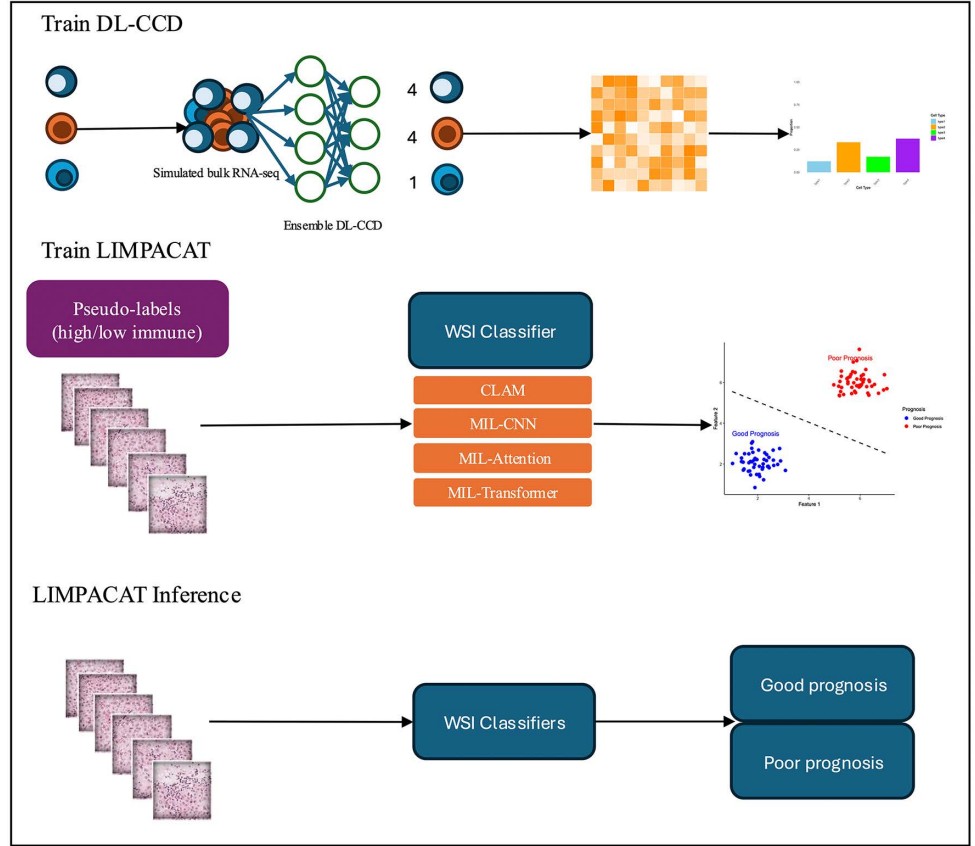

**Fig 1. Immune cell compositions were initially inferred from HCC scRNA-seq data via a DL-CCD model, generating pseudo-labels for high vs. low immune levels.** These labels provided weak supervision to train MIL–based WSI classifiers (CLAM, MIL-CNN, MIL-Attention, MIL-Transformer). At inference, trained classifiers directly predicted immune infiltration from pathology slides, enabling survival stratification into favorable vs. poor prognosis groups.

Quality control (QC) was applied at the single-cell level to exclude potential artifacts and low-quality populations. Cells were retained if the number of detected genes (nFeature_RNA) ranged from 250 to 2,500, thereby minimizing potential inclusion of empty droplets, doublets, or highly complex multiplets. Additionally, cells with mitochondrial gene expression exceeding 5% of total counts (PercentageFeatureSet with pattern = "^MT-") were excluded, as elevated mitochondrial content is typically indicative of apoptotic or damaged cells. These QC thresholds were consistently applied across all samples before downstream processing.

Following QC, data normalization and batch correction were performed using three distinct strategies implemented in Seurat. First, global-scaling normalization with the NormalizeData function (method = "LogNormalize") rescaled gene expression by total counts per cell [15]. Second, CCA–based integration, implemented with the FindIntegrationAnchors and IntegrateData functions, aligned samples within a shared low-dimensional space [16]. Third, a variance-stabilizing transformation within the SCTransform function directly modeled sequencing depth and technical noise [15]. Each normalization strategy was subsequently applied for clustering and dimensionality reduction to assess consistency and robustness across methods.

Real-world HCC data were obtained from the TCGA-LIHC cohort available on the Genomic Data Commons Data Portal, including gene expression profiles, diagnostic WSIs, and corresponding clinical records. Gene expression data

were preprocessed using the upper-quartile normalization. Clinical and survival records were linked to WSIs through case identifiers. Following quality control, 326 WSIs from 312 unique patients were retained, with some patients contributing multiple slides. To minimize potential clustering bias, downstream analyses were performed at the patient level by aggregating slides from the same case. S1 Table shows the baseline cohort characteristics—including age at diagnosis, clinical risk factors, primary diagnosis, follow-up duration, and WSI distribution per patient. The median follow-up time was 26.4 months, and most patients contributed only a single diagnostic WSI (mean = 1.04, range: 1–2).

## Evaluation of single-cell RNA sequencing normalization

To assess clustering consistency and the effectiveness of batch effect correction, adjusted rand index (ARI) comparisons were performed across normalization strategies [17,28–30]. The ARI, calculated as quantifies the alignment between clustering results and reference groupings, while adjusting for chance alignment. Here, $n$ denotes the total number of data points (e.g., cells), $n_{ij}$ represents the number of data points shared between cluster $i$ in one clustering and cluster j in the reference clustering. The sums $a_i$ and $b_j$ represent the total number of points in cluster $i$ and cluster $j$, respectively. We evaluated alignment between cell type annotations, clustering results, and liver region-specific batch effects derived from multiple patients and liver regions within the dataset.

1. A–C (Annotation vs. Clustering): Cell annotation was performed using SingleR, which classified cells into biologically relevant types based on reference datasets. Clustering was conducted using the unsupervised approach by Seurat. The ARI was calculated to quantify the alignment between SingleR annotations and Seurat clusters. A high ARI score indicates that Seurat clustering effectively identifies biologically distinct cell subpopulations.

2. B–C (Batch vs. Clustering): To evaluate the influence of liver region-specific batch effects on clustering, ARI was computed between liver region batches and Seurat clustering results. In this context, a low ARI score is desirable, reflecting that clustering is driven by biological variation rather than technical batch effects, demonstrating effective batch effect correction.

3. A–B (Annotation vs. Batch): ARI was also calculated between SingleR-derived cell annotations and liver region batches to assess the distribution of annotated cell types across regions. A low ARI score in this comparison indicates that annotated cell types are consistently distributed across liver regions without confounding batch effects, reinforcing the stability of annotations across spatially distinct regions.

## Immune cell prediction model development using single-cell RNA sequencing

Cell types were annotated using the SingleR package [31], which matches scRNA-seq data with reference profiles to identify specific immune cell types. Based on these annotations, we developed an ensemble deep neural network (ensemble-DNN) model to predict immune cell composition from gene expression data, following the framework established in our previous study [8]. Training data were generated by constructing pseudo-bulk RNA-seq samples through aggregation of scRNA-seq profiles according to predefined cell-type proportions, following the framework established in a previous study [8]. The methodology and validation of this approach, including comparisons with widely used deconvolution methods, have been detailed in a previous study.

## Digital pathology image analysis

WSIs were processed using MONAI's WSIReader with the cuCIM backend (uint8) at pyramid level 1 and divided into 224 × 224 pixel patches. During training, RandGridPatchd was applied to sample 44 instances per slide, with sort_fn = "min" prioritizing tissue-rich regions. For validation, GridPatchd generated a deterministic grid of tiles. The Split-Dimd operation was used to separate the patch dimension into individual instances. Data augmentation during training

   

included random horizontal/vertical flips and 90° rotations (RandFlipd, RandRotate90d), whereas validation relied solely on deterministic preprocessing. Pixel intensities were linearly rescaled from [0, 255] to a normalized range before tensor conversion.

Slide-level supervision followed a weakly supervised MIL framework. Ordinal targets were encoded with a cumulative multi-hot representation through a custom LabelEncodeIntegerGraded transform, where all positions up to the label index were set to 1. The loss function was computed with BCEWithLogitsLoss over the encoded vector.

We employed MONAI's milmodel.MILModel with an ImageNet-pretrained ResNet-50 backbone for patch feature extraction, and evaluated two aggregation variants: gated-attention MIL (ATT) and transformer-augmented attention MIL (ATT_TRANS). In ATT, instance features were projected and weighted through a gated-attention module before weight pooled into a slide-level representation. In ATT_TRANS, the attention-pooled sequence was further modeled by a transformer encoder to capture inter-patch dependencies prior to classification. To manage memory during validation, slide yielding more than 44 patches was randomly subsampled to 44 instances before inference, and the final prediction was based on slide-level logits.

Models were trained in PyTorch using AdamW (learning rate $= 3 \times 10^{-5}$) with a cosine annealing scheduler. For ATT_TRANS, the transformer block used a lower learning rate ($6 \times 10^{-6}$) and weight decay of 0.1 through a separate optimizer parameter group, while the backbone, attention, and classification head shared the base learning rate. Mixed-precision training (AMP) was enabled, with mini-batches containing up to four WSIs. Early stopping was applied based on validation performance, and model evaluation was reported using accuracy and quadratic-weighted Cohen's kappa (QWK). All experiments were conducted on NVIDIA V100.

For comparison, we implemented CLAM following the published guidelines and codebase of the author. The model was trained with its original ImageNet-pretrained ResNet-50 feature extractor and attention-based MIL pooling, applied directly to the same WSI dataset. This served as a consistent baseline for evaluating LIMPACAT and the proposed ATT/ATT_TRANS variants under equivalent conditions.

## Evaluation methods

To evaluate the generalizability and robustness of the proposed CCD model, external validation was performed using real-world peripheral blood mononuclear cell (PBMC) bulk RNA-seq samples from the GSE107011 dataset. This dataset provides bulk RNA-seq expression profiles from sorted immune cell populations with well-characterized compositions.

Predicted cell-type proportions were compared to reference proportions using the Pearson correlation coefficient (PCC), a standard metric for evaluating deconvolution performance in both simulated and real bulk RNA-seq settings. Higher PCC values indicate stronger concordance between inferred and reference cell compositions. This methodology is consistent with evaluation strategies applied in previous benchmarking studies of cell-type deconvolution models [8–10,32].

To evaluate whether WSIs could predict immune microenvironment patterns, we used predicted high vs. low immune cell groupings as pseudo-labels for training and validating weakly supervised WSI classifiers. Classification accuracy was defined as the proportion of correctly predicted labels compared to those derived from scRNA-seq immune groups. Ninety-five percent confidence intervals (CIs) for accuracy were calculated from the sample mean and standard error, using the t-distribution with n-1 degrees of freedom.

## Survival analysis

Overall survival was assessed using the Kaplan–Meier method, and differences between survival distributions were tested with the log-rank test. For each immune cell type, patients were stratified into "high" and "low" groups based on the median estimated immune cell fraction, derived from a scRNA-seq-informed deconvolution model applied to bulk RNA-seq profiles. This binary stratification enabled univariate survival comparisons, with statistical significance defined as $p < 0.05$.

## Results

### Gene count and quality control metrics of the GSE189903 dataset

The GSE189903 dataset comprises 20 HCC samples, encompassing ~737,280 cells and 11,632 shared genes. Before filtering, per-cell gene counts ranged from ~12,000 to >20,000, indicating substantial heterogeneity (Fig 2A). After quality control, the number of retained cells varied across samples, from < 5,000 to nearly 15,000 (Fig 2B). Despite this reduction, most samples retained 15,000–20,000 gene features, ensuring sufficient expression coverage for downstream analyses (Fig 2C). The proportion of retained cells per sample ranged from 5% to 25% of the original counts, reflecting stringent filtering while preserving tumor microenvironment representation (Fig 2D). Correlation analysis among filtering metrics further confirmed procedural consistency (S1 Fig).

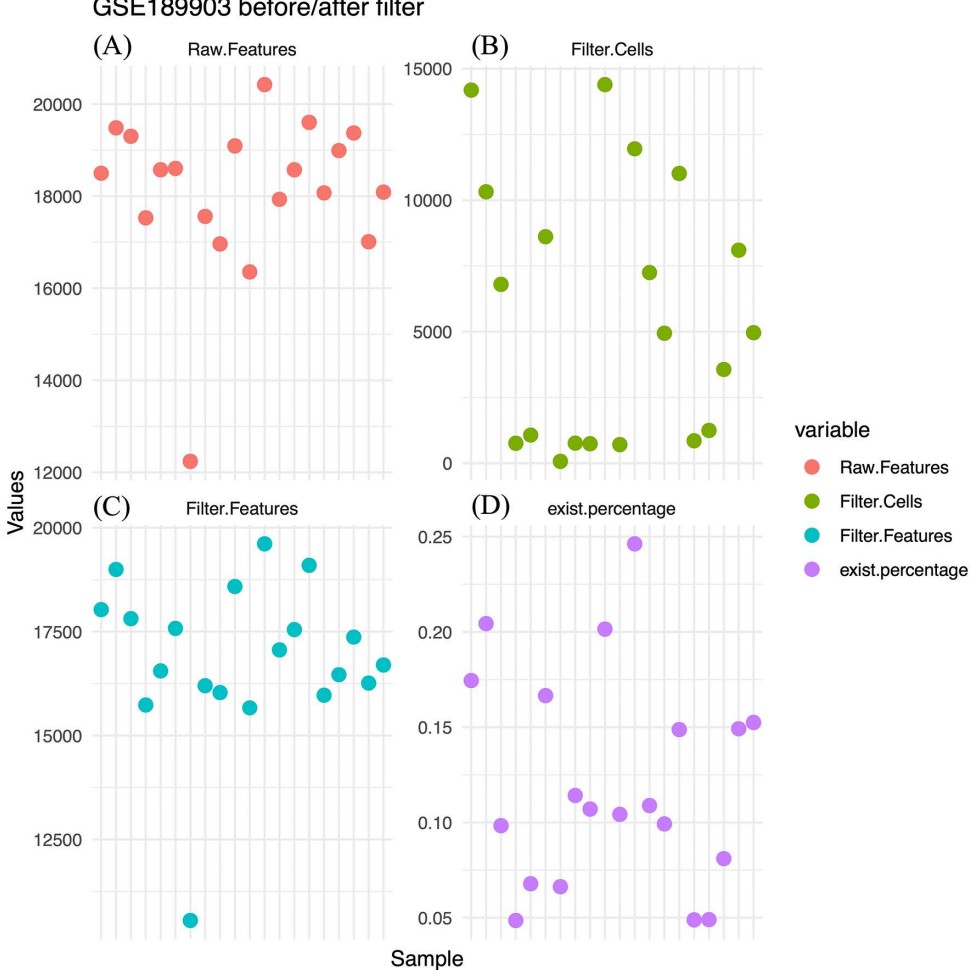

**Fig 2. Distribution of features and cells before and after filtering in the GSE189903 dataset. (A)** Distribution of gene counts per cell prior to quality filtering. **(B)** Number of cells retained after quality control in each of the 20 HCC samples. **(C)** Number of detected genes per sample after filtering. **(D)** Proportion of retained cells relative to the original cell count in each sample.

## Correlation analysis and data quality evaluation

The distribution of nFeature values highlighted heterogeneity in gene detection per cell, with most cells containing < 100 genes and a substantial subset ranging from 250–2,500 (S2 Fig). Following filtering, mitochondrial gene content (mt percentage) and nCount distributions were assessed for each sample to evaluate sequencing quality and depth.

Across samples, mt percentages exhibited modest variability, with means of 1.42–3.38, medians generally concordant with means, and standard deviations of 0.85–1.38, indicating moderate dispersion. Minimum mt percentages were consistently 0, while maximum values approached 5, reflecting the applied filtering threshold.

Cell retention varied substantially across samples, ranging from 69 cells in sample 2HN to 14,183 cells in sample 1HB (S3A in S3 Fig and S2 Table). Similarly, nCount distributions exhibited wide differences in sequencing depth, with mean values ranging from 879 to 4,204, and individual cells spanning ~280 to >40,000. These metrics highlight heterogeneity in sequencing quality and cell representation (S3B in S3 Fig and S3 Table). Overall, filtering retained approximately 10–25% of the original cells, yielding final counts from dozens to tens of thousands across samples samples (S3 Fig and S4 Table).

To further evaluate data quality, correlation analyses were performed. nCount and mitochondrial percentage exhibited weak correlation, indicating that mitochondrial content was largely independent of sequencing depth (S4 Fig). In contrast, nCount and nFeature showed a strong positive correlation, consistent with the expected relationship between sequencing depth and gene detection, thereby confirming the reliability of retained cells (S5 Fig).

## Comparison of single-cell RNA sequencing normalization methods

To integrate data across multiple samples and eliminate batch effects, three normalization strategies were compared: log normalization, CCA, and SCTtransform. Each method was evaluated for clustering consistency and sample distribution using UMAP and ARI.

For log normalization, UMAP identified 25 clusters (Fig 3 A). Comparison of clustering consistency between the 20 sample types and these 25 clusters yielded an ARI score of 0.168 (Fig 3 B). In contrast, CCA normalization produced 28 clusters with a significantly lower ARI score of 0.04 (S6 Fig). SCTtransform normalization generated 23 clusters, with an ARI score of 0.043 (S7 Fig). This method showed an intermediate performance between log normalization and CCA. Log normalization achieved the highest ARI score in most comparisons, highlighted in yellow, indicating its superior clustering consistency (S5 Table).

## Comparison of cluster correlation

To evaluate the effect of log normalization, CCA, and SCTransform on data structure, we examined nFeature and nCount distributions across samples, as well as sample-to-sample correlation patterns.

Under log normalization, nFeature and nCount distributions exhibited broader ranges across samples, reflecting preserved heterogeneity in gene expression (Fig 4A). Importantly, this heterogeneity captures biologically relevant diversity among cells, which may be masked by stronger batch correction methods. The corresponding sample-to-sample correlation heatmap showed moderate alignment (Fig 4B), indicating that log normalization avoids over-standardization and thus preserves meaningful biological variability. This balance makes log normalization particularly suitable as the foundation for our CCD framework.

In contrast, CCA and SCTransform produced more concentrated nFeature and nCount distributions (S8A-B in S8 Figs) and high cross-sample correlations (S8C-D in S8 Figs), highlighting their strong batch effect reduction. However, this high degree of alignment may come at the cost of reduced biological resolution, potentially limiting their utility when subtle immune-cell differences are of interest.

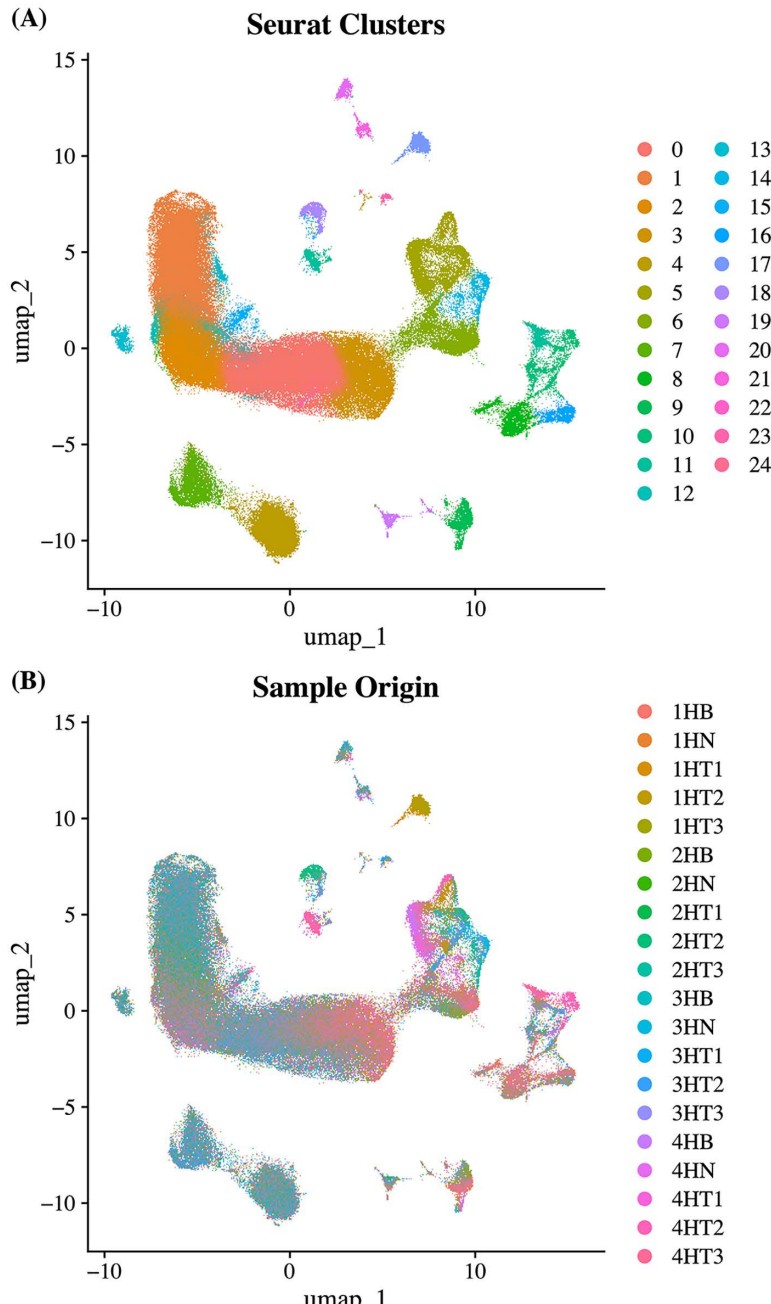

**Fig 3. UMAP clustering of scRNA-seq data following log normalization. (A)** UMAP plot showing 25 Seurat-defined clusters, capturing cellular diversity within the dataset. **(B)** UMAP plot with cells colored by sample identity, illustrating cluster distribution across samples.

## Single-cell type annotation

To classify cell types in HCC samples, three normalization methods were applied for single-cell annotation. Across all methods, T cells represented the most abundant population within the dataset, followed by monocytes/NK cells and cancer cells. Conversely, neutrophils, hematopoietic stem cells (HSC), and Pre-B cells were consistently observed at

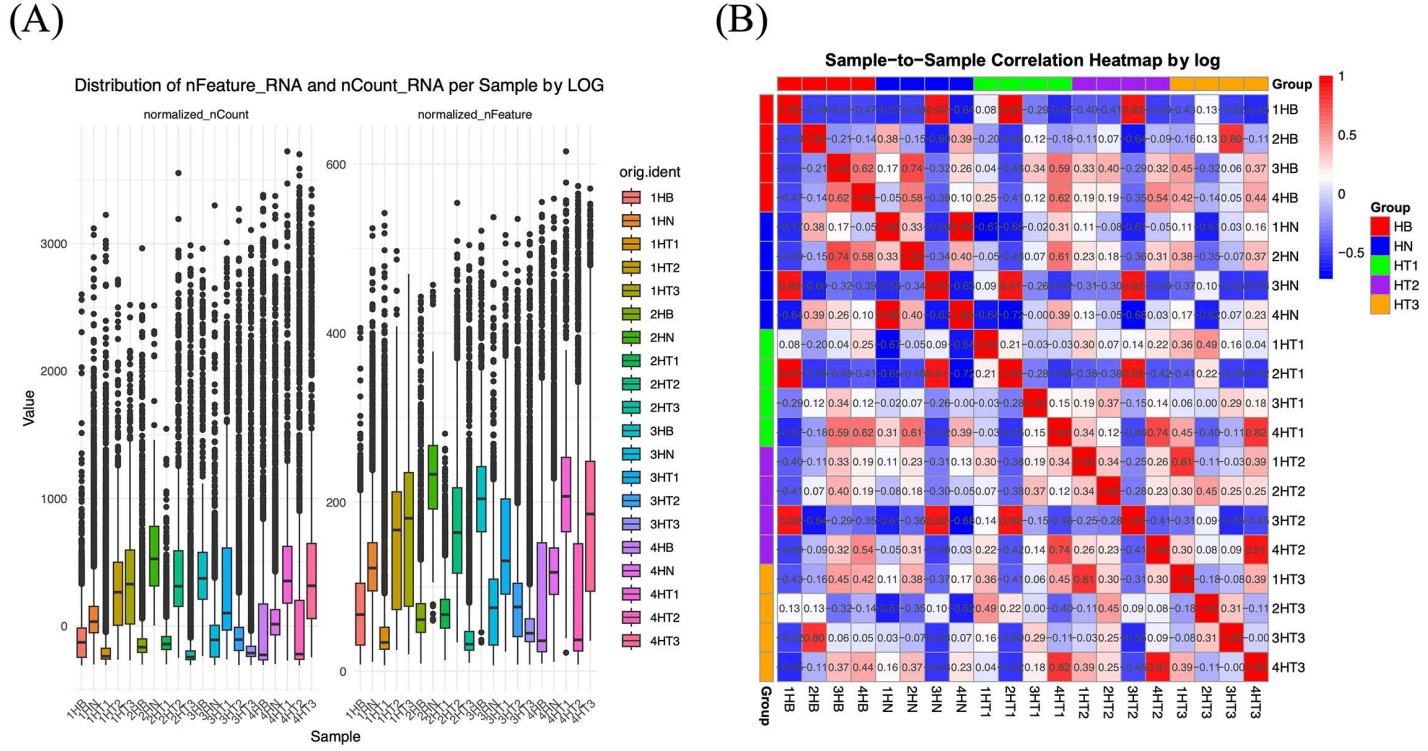

**Fig 4. nFeature and nCount distributions. (A)** Boxplots showing balanced feature and count distributions across samples **(B)** Sample-to-sample correlation heatmap for log normalization, indicating high consistency.

lower frequencies (S6-S8 Tables). Under log normalization, UMAP clustering revealed the distribution of annotated cell types, highlighting the predominance of T cells (Fig 5A). Table S6 presents a quantitative summary of each cell type, with T cells constituting the largest portion, totaling 94,602 cells. Monocytes (3,979 cells) and NK cells (4,576 cells) were also relatively abundant, whereas neutrophils and specific HSC subtypes were minimally represented. Using CCA normalization, UMAP visualization identified 12 distinct cell types, with T cells remaining predominant, totaling 97,005 cells (S9A in S9 Fig). S7 Table shows that monocytes (3,299 cells) and NK cells (2,615 cells) followed in abundance, while neutrophils and HSC subtypes again exhibited low frequencies. SCTtransform normalization yielded comparable results, with T cells as the predominant population (96,693 cells) according to UMAP analysis (S9B in S9 Fig). S8 Table presents the cell counts, showing monocytes (4,351 cells) and NK cells (2,564 cells) with distributions consistent across normalization methods. Neutrophils and HSCs remained sparse, highlighting a recurrent trend across methods. Fig 5 B presents a comparative summary of cell type distributions across the three normalization methods. These findings highlight the robustness of the SingleR package in identifying major immune cell types within HCC samples, alongside its consistency in detecting rare cell types, such as neutrophils and specific HSC subtypes, across normalization methods.

## Cell composition deconvolution

To develop a model for analyzing digital pathology slides and predicting immune cell levels associated with patient prognosis, the TCGA-LIHC dataset was used in this study. Given that this dataset does not include direct measurements of immune cell composition, the first step involved inferring immune cell composition for each HCC sample. Simulated bulk

(A)

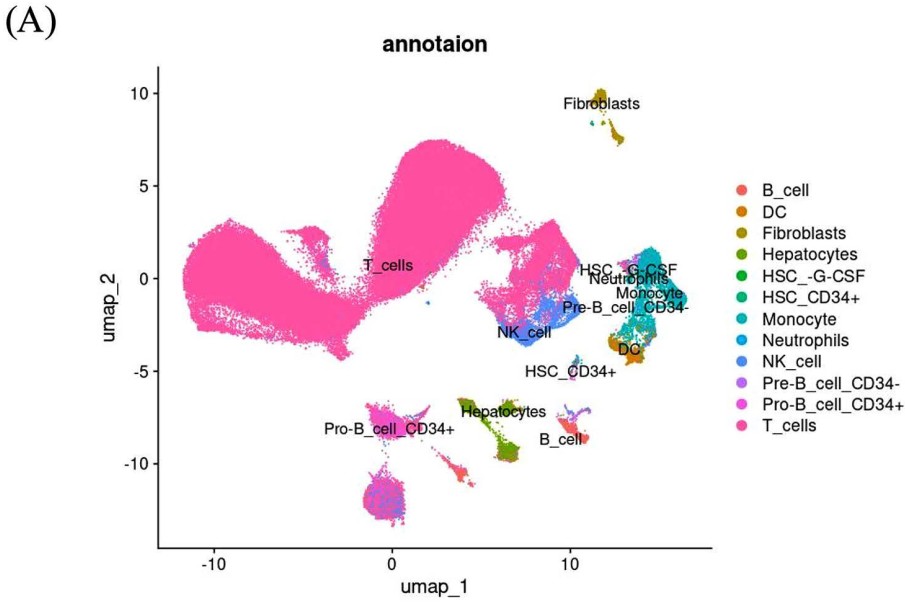

(B)

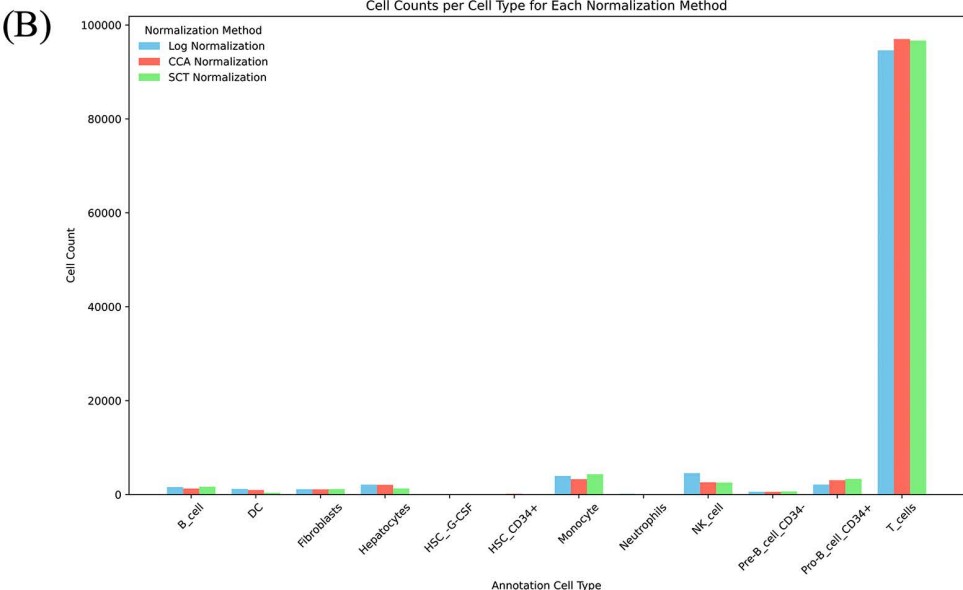

**Fig 5. Influence of three normalization methods on immune cell annotation and distribution in the HCC single-cell transcriptomic dataset.**
**(A)** UMAP visualization of cell type annotations following log normalization, illustrating the distribution of major immune cell types, including T cells and monocytes, along with less abundant populations such as neutrophils and specific HSC subtypes. **(B)** Bar chart comparing cell type distributions across the three normalization methods: log, CCA, and SCT.

RNA-seq expression profiles were generated from liver-specific scRNA-seq data and using three different normalization methods, with immune cell counts serving as markers. An ensemble-DNN cell deconvolution model was trained on 8,000 simulated datasets and validated on 2,000 datasets, demonstrating stable performance without signs of overfitting in accuracy or error rate (S10A in S10 Fig).

Subsequently, the liver scRNA-seq ensemble-DNN model was applied to estimate the cell composition in real bulk RNA-seq profiles from HCC samples. Among the three normalization methods, CCA- and SCT-normalized data resulted in similar immune cell composition distributions in LIHC samples. In contrast, log normalization achieved the highest clustering consistency and preserved more biological variability across samples, which is essential for capturing tumor microenvironment heterogeneity (S10B-C in S10 Figs). This higher resolution facilitated better identification of biologically relevant subpopulations while maintaining consistent annotations across datasets. Based on these results, and after integrating the ARI and correlation analyses (S10B-D in S10 Figsand S5 Table), the log-normalized CCD model was selected for downstream survival analysis. The survival analysis revealed a positive association between higher levels of B cells, NK cells, and monocytes and longer survival times, while increased CD34$^+$B cells were associated with poorer prognosis (S11 Fig and S9 Table).

To further validate the CCD model, it was applied to real PBMC bulk RNA-seq samples from the GSE107011 dataset using a manually curated cell type correspondence table (S10 Table). The inferred cell type proportions were compared with expected distributions derived from sorted immune populations. The model demonstrated strong concordance with ground-truth proportions, achieving Pearson correlation coefficients ranging from 0.84 to 0.98 across most samples (S12 Fig). These findings highlight the ability of the model to accurately recover diverse immune cell populations from real PBMC samples, including commonly investigated subtypes such as regulatory T cells, effector memory CD8+T cells, and memory B cells.

### Immune cell composition estimates from whole-slide images

In image-based survival classification using CCD-derived stratification, the performance of CLAM, CNN, ATT, and ATT_TRANS was evaluated across B cell, NK cell, monocyte, and CD34$^+$B cell signatures. CLAM consistently demonstrated the lowest validation accuracies (36–43%), serving as the baseline reference. Both CNN and ATT improved performance across all signatures, with ATT generally achieving higher accuracy than CNN. The ATT outperformed all other models in lymphoid cell signatures, achieving 70% accuracy for B cells and 79% for NK cells. In contrast, the ATT_TRANS achieved the highest accuracy in myeloid and precursor cell signatures, reaching 75% for both monocytes and CD34$^+$B cells. In these instances, the lower bounds of ATT_TRANS performance exceeded the upper bounds of the other models, highlighting its relative advantage. Overall, attention-based architectures outperformed CLAM and CNN, with ATT favored for lymphoid cell prediction and ATT_TRANS for monocyte-related tasks (Table 1). We further applied the ATT_TRANS model to WSI data to stratify patients into high and low groups for the four immune cell types, and Kaplan–Meier analysis yielded results consistent with those obtained using the CCD-based predictions (S11 Fig and S13 Fig).

## Discussion

LIMPACAT successfully predicted immune cell abundances by applying MIL attention models to WSIs. The results showed a positive correlation between the abundance of B and NK cells and improved survival outcomes in patients with HCC. This finding is consistent with those of previous studies highlighting the roles of B and NK cells in HCC immunity. Zhao et al. report distinct B-cell subsets within the HCC microenvironment: activated B cells demonstrate increased metabolic activity associated with enhanced effector functions, while exhausted B cells show reduced functionality, indicating the diverse roles of B-cell subsets in tumor suppression and immune modulation [33]. Additionally, Qin et al. show that high infiltration of cytokine-secreting B cells with antigen presentation capacity is associated with improved patient survival, supporting the immunoregulatory and tumor-suppressive roles of B cells in HCC [34]. Similarly, Zou et al. emphasize the role of B cells in shaping the immune microenvironment through T cell activation and antibody production, collectively facilitating antitumor immunity and correlating with favorable patient prognosis [35]. Consistent with these findings, our results showed that increased B-cell abundance in the liver tumor microenvironment was associated with improved survival outcomes.

**Table 1. Validation Accuracies of CLAM, CNN, ATT, and ATT_TRANS across B cell, CD34⁺ B cell, monocyte, and NK cell signatures.**

|  | CLAM (%) | CNN (%) | ATT (%) | ATT_TRANS (%) |
|---|---|---|---|---|
| B cell (Train) | 46.7 [30.2, 63.9] | 78.1 [73.4, 83.3] | 97.1 [95.6, 98.6] | 86.2 [81.7, 90.3] |
| B cell (Validation) | 43.3 [27.4, 60.8] | 64.7 [62.7, 66.5] | 69.7 [68.8, 70.7] | 62.6 [60.5, 64.7] |
| NK cell (Train) | 46.7 [30.2, 63.9] | 95.8 [93.6, 98.0] | 98.1 [97.1, 99.1] | 91.7 [87.2, 96.0] |
| NK cell (Validation) | 40.0 [24.6, 57.7] | 76.6 [75.3, 78.0] | 78.8 [77.9, 80.0] | 73.7 [72.7, 75.0] |
| Monocyte cell (Train) | 43.3 [27.4, 60.8] | 96.3 [95.2, 97.4] | 91.3 [90.2, 92.2] | 92.2 [88.1, 96.2] |
| Monocyte cell (Validation) | 36.7 [21.9, 54.5] | 58.6 [58.0, 59.2] | 70.0 [68.8, 70.1] | 74.8 [73.7, 75.7] |
| CD34⁺B cell (Train) | 46.7 [30.2, 63.9] | 92.7 [90.0, 95.1] | 97.5 [98.2, 96.8] | 85.1 [84.5, 85.9] |
| CD34⁺B cell (Validation) | 40.0 [24.6, 57.7] | 62.7 [62.0, 64.1] | 63.5 [63.2, 63.8] | 74.8 [72.9, 77.4] |

Similarly, NK cells emerged as a crucial component of the immune response in this study, demonstrating significant contributions to antitumor immunity in HCC. This finding is consistent with those of Xi et al. and Sajid et al., highlighting the ability of NK cells to mediate cytotoxicity both directly—through granzyme and perforin release—and indirectly, via cytokine secretion, such as IFN-γ, which activates broader immune responses against tumor cells [36,37]. In this study, the presence of NK cells was positively associated with patient survival, supporting previous findings and indicating their potential in improving HCC prognosis through enhanced immune surveillance.

In this study, an attention-based transformer model within an MIL framework was used to selectively focus on immune-rich regions in WSIs, thereby enhancing prognostic predictions through accurate stratification of immune cell levels within the tumor microenvironment. In contrast to previous studies, this approach was used to emphasize immune cell composition as a primary prognostic factor. For example, Feng et al. developed a sliding-attention transformer model to predict T cell receptor–antigen interactions, facilitating neoantigen discovery for immunotherapy. However, the study emphasizes molecular-level immune responses rather than immune stratification within the tumor microenvironment [38]. Similarly, Duanmu et al. applied spatial attention mechanisms to predict treatment response in breast cancer pathology, focusing on therapy effectiveness rather than immune cell abundance [39].

Additionally, Xiong et al. developed a hierarchical attention-guided MIL framework for WSI classification, employing attention mechanisms to identify critical regions within images. However, the model primarily targets general cancer classification without addressing immune cell stratification, a central element of this study [40]. Mahmood et al. proposed a convolution–transformer hybrid model integrating adaptive convolution and dynamic attention mechanisms to capture fine-grained features in renal cell carcinoma images. While effective for classification, their approach differs from our immune-focused framework, specifically designed for prognostic applications [41].

In contrast to the CLAM model developed by Lu et al.—a weakly supervised MIL framework applying attention mechanisms for WSI classification—our approach adopts a different strategy. Although CLAM effectively identifies high-diagnostic-value regions with minimal labeling [19], its primary focus is on multiclass pathology classification, rather than immune c1ll deconvolution for direct prognostic applications. In this study, CLAM achieved only 50% accuracy in classifying immune cell levels within the tumor microenvironment, highlighting its limitations for this specific task. In contrast, the LIMPACAT model, explicitly designed for immune cell deconvolution, reached approximately 80% accuracy, representing a significant improvement in predicting patient immune infiltration of B and NK cells.

In the model structure, transformer-based advancements in WSI analysis primarily focus on developing large-scale self-supervised encoders and efficient strategies for aggregating long sequences. Foundation models such as UNI [42], Virchow [43], and Prov-GigaPath [44] demonstrate strong cross-organ transfer capabilities, indicating that substituting our current encoder with a pretrained Vision Transformer could further enhance generalization. At the bag-aggregation stage, hierarchical and sequence-efficient architectures, such as HVTSurv [45], TransMIL [46], and MambaMIL [47], emphasize the importance of capturing multi-scale contextual information and modeling ultra-long token sequences, particularly relevant for immune cell inference from H&E-stained slides. More directly aligned with our task, frameworks such as HistoTME focus on inferring tumor microenvironment composition from WSIs, linking these estimates to immunotherapy outcomes [48]. In contrast, LIMPACAT integrates weak supervision with single-cell–derived deconvolution: scRNA-seq is used to generate biologically grounded immune labels, while MIL models—including a transformer-enhanced ATT_TRANS variant—are trained to infer immune composition from WSIs.

Recent studies demonstrate the feasibility of validating computational predictions of immune infiltration using immunohistochemical (IHC) assays. For example, Zhang et al. applied a deep learning framework (TILDL) to quantify tumor-infiltrating lymphocytes in nasopharyngeal carcinoma, reporting strong concordance between computationally derived TIL percentages and IHC-based measurements of CD3+, CD8+, and CD20+ cells. The computational approach outperformed IHC in prognostic accuracy [49]. Similarly, several studies show that CNN-based lymphocyte maps strongly correlate with IHC-derived immune cell densities across various tumor types, highlighting the reliability of image-based prediction models when validated against experimental ground truths [50]. These findings indirectly support the validity of our approach, as LIMPACAT applies similar deep learning and deconvolution principles to infer immune cell composition from WSIs and scRNA-seq integration. Future studies should incorporate orthogonal validation, particularly IHC or multiplex immunofluorescence (mIF) in HCC cohorts, to further substantiate and extend these findings.

However, a key limitation of this study is the lack of direct experimental validation—such as flow cytometry, IHC, mIF, or spatial transcriptomics—to confirm the predicted immune cell abundances. While the findings are consistent with those of previous studies and are indirectly supported by existing literature, future studies should validate these findings in independent cohorts using orthogonal experimental approaches, such as IHC or mIF, to enhance the robustness, interpretability, and generalizability of the model.

Furthermore, while this study demonstrates the feasibility of predicting immune cell composition from WSIs using weak supervision grounded in single-cell deconvolution, some limitations remain. Patients were stratified into high- and low-infiltration groups based on median estimated immune cell fractions and group differences evaluated using Kaplan–Meier curves and log-rank tests. However, the concordance index (C-index) was not calculated, and immune scores were not analyzed as continuous predictors. This approach was chosen to maintain interpretability and reflect the dichotomous nature of the weak supervision labels derived from scRNA-seq. Future studies should incorporate continuous immune predictors, multivariate Cox proportional hazards models adjusted for clinical covariates, and C-index-based evaluation to provide a more comprehensive assessment of prognostic relevance.

## Conclusion

The LIMPACAT framework developed in this study demonstrates the feasibility of integrating scRNA-seq and WSI data to accurately predict immune cell composition and patient prognosis in HCC. The findings reveal the value of comprehensive tumor microenvironment analyses, highlighting B cells, CD34+ B cells, monocytes, and NK cells as key prognostic indicators. Although challenges related to scRNA-seq annotation, model performance, and generalizability remain, a key limitation of this study is the lack of direct biological validation (e.g., IHC or mIF) to confirm the predicted immune cell abundances. Future studies incorporating these experimental validations will be essential for establishing the biological accuracy and translational relevance of LIMPACAT. Overall, this framework offers valuable insights into the cancer immune microenvironment and presents innovative strategies for integrating multi-omics data with image analysis, highlighting the potential of these models in precision oncology.

## Supporting information

**S1 Fig. Correlation matrix showing relationships among filtering metrics (Raw Features, Filtered Cells, Filtered Features, Retained Cell Percentage) in the GSE189903 dataset.** Positive correlations confirm the effectiveness of the filtering process.
(PDF)

**S2 Fig. Filtered Cell Population with nFeature Counts Between 250 and 2500.**
(PDF)

**S3 Fig. Distribution of mitochondrial gene percentage (mt percent) and nCount across samples post-filtering.** (A) shows the mt percent for each sample, reflecting cell viability and quality, with higher percentages indicating potential cellular stress or damage. (B) displays the nCount distribution, representing the sequencing depth across samples, which captures gene expression patterns specific to different cell types or states.
(PDF)

**S4 Fig. Correlation between nCount and mitochondrial gene percentage (mt percent) across samples.** The weak correlation indicates that mitochondrial gene content is independent of sequencing depth, suggesting minimal impact of sequencing depth on mitochondrial percentage.
(PDF)

**S5 Fig. Correlation between nCount and nFeature across samples.** The strong positive correlation confirms that cells retained after filtering have consistent sequencing depth and gene detection, supporting the reliability of high-quality data for downstream analysis.
(PDF)

**S6 Fig. UMAP clustering of scRNA-seq data post-CCA normalization.** (A) UMAP plot showing the distribution of 28 clusters, highlighting additional cell state distinctions. (B) UMAP plot colored by sample identity, revealing the distribution of cells across clusters. Lower ARI score suggests less consistency compared to log normalization.
(PDF)

**S7 Fig. UMAP clustering of scRNA-seq data by sct normalization.** (A) UMAP plot with 23 defined clusters, indicating preserved cell type distinctions. (B) UMAP plot by sample identity, showing the spread of cells across clusters. The ARI score indicates moderate clustering consistency and successful batch effect mitigation.
(PDF)

**S8 Fig. nFeature and nCount distributions and sample-to-sample correlations for sct normalization (A, C) and CCA (B, D).** The boxplots show the distributions of nFeature and nCount across samples, and the heatmaps display sample-to-sample correlation patterns under each normalization method.
(PDF)

**S9 Fig. Cell type annotations after clustering with sct normalization (A) and CCA (B).** UMAP visualizations show identified cell types, based on annotation results following each normalization method.
(PDF)

**S10 Fig. Summary of cell composition deconvolution model performance, normalization comparisons.** and survival analysis. (A) Training and validation accuracy/error rates for the cell deconvolution model show no overfitting. (B) ARI comparison of immune cell composition predictions across normalization methods (LOG, CCA, SCT) for LIHC samples. (C) Correlation of immune cell composition and survival times across normalization methods, with CCA and SCT showing consistency. (D) Immune cell composition distributions in LIHC samples, showing similarity between CCA and SCT.
(PDF)

**S11 Fig. Survival analysis indicates a positive association between higher levels of NK cells, B cells, and mono-cytes with longer survival times, whereas a negative association is observed with higher levels of CD34+B cells.**
(PDF)

**S12 Fig. Validation of the CCD model on real PBMC bulk RNA-seq samples from GSE107011.** Pearson correlation coef-ficients between model-inferred cell-type proportions and ground truth proportions derived from sorted PBMC populations.
(PDF)

**S13 Fig. Survival analysis based on WSI-predicted immune levels.**
(PDF)

**S1 Table. Summary of the TCGA-LIHC cohort.**
(XLSX)

**S2 Table. Filter_mt_persample.**
(XLSX)

**S3 Table. Filter_count_persample.**
(XLSX)

**S4 Table. Filtered cell counts and mitochondrial gene proportion after quality control.**
(XLSX)

**S5 Table. ARI between annotation, batch and clustering.**
(XLSX)

**S6 Table. Cell type distribution and counts after log normalization.** The table shows the number of cells for each identified cell type across samples, classified based on log normalization.
(XLSX)

**S7 Table. Cell type distribution and counts after CCA normalization.** The table provides the cell counts for each iden-tified cell type across samples, classified following CCA normalization.
(XLSX)

**S8 Table. Cell type distribution and counts after SCT normalization.** This table displays the number of cells for each identified cell type across samples, categorized based on SCT normalization.
(XLSX)

**S9 Table. Log-Rank Test.**
(XLSX)

**S10 Table. LIMPACAT (Target Cell Type).**
(XLSX)

## Acknowledgments

The authors thank the National Center for High-Performance Computing (NCHC), Taiwan, for computational resources.

## Author contributions

**Conceptualization:** Yen-Jung Chiu.

**Data curation:** Yen-Jung Chiu.

**Formal analysis:** Yen-Jung Chiu.

**Funding acquisition:** Yen-Jung Chiu.

**Investigation:** Yen-Jung Chiu.

**Methodology:** Yen-Jung Chiu.

**Project administration:** Yen-Jung Chiu.

**Resources:** Yen-Jung Chiu.

**Software:** Yen-Jung Chiu.

**Supervision:** Yen-Jung Chiu.

**Validation:** Yen-Jung Chiu.

**Visualization:** Yen-Jung Chiu.

**Writing – original draft:** Yen-Jung Chiu.

**Writing – review & editing:** Yen-Jung Chiu.

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
