## [Decision Letter · Decision Letter 0]

17 Aug 2025

Dear Dr. Chiu,

Thank you for submitting your manuscript to PLOS ONE. After careful consideration, we feel that it has merit but does not fully meet PLOS ONE’s publication criteria as it currently stands. Therefore, we invite you to submit a revised version of the manuscript that addresses the points raised during the review process.

We look forward to receiving your revised manuscript.

Kind regards,

Li Yang, M.D.

Academic Editor

PLOS ONE

Journal Requirements:

This work was supported by the grants from the National Science and Technology Council (NSTC).

This research was supported by grants from the National Science and Technology Council (NSTC), Taiwan (NSTC 112-2222-E-130-003- and NSTC 113-2221-E-130-005-MY3), which assisted in the acquisition of computational facilities essential for this study.

No available

5. In the online submission form, you indicated that your data will be submitted to a repository upon acceptance. We strongly recommend all authors deposit their data before acceptance, as the process can be lengthy and hold up publication timelines. Please note that, though access restrictions are acceptable now, your entire minimal dataset will need to be made freely accessible if your manuscript is accepted for publication. This policy applies to all data except where public deposition would breach compliance with the protocol approved by your research ethics board. If you are unable to adhere to our open data policy, please kindly revise your statement to explain your reasoning and we will seek the editor's input on an exemption.

6. Please amend your list of authors on the manuscript to ensure that each author is linked to an affiliation. Authors’ affiliations should reflect the institution where the work was done (if authors moved subsequently, you can also list the new affiliation stating “current affiliation:….” as necessary).

7. Please upload a new copy of Figure 1 as the detail is not clear. Please follow the link for more information: https://blogs.plos.org/plos/2019/06/looking-good-tips-for-creating-your-plos-figures-graphics/"" https://blogs.plos.org/plos/2019/06/looking-good-tips-for-creating-your-plos-figures-graphics/

Additional Editor Comments:

Thanks for submitting your work to PLOS ONE. Your manuscript has now been assessed by our editorial team and external peer experts. While they found it interesting, you will see that they have raised many serious problems and are advising that you revise your manuscript thoroughly. At the same time, please submit the point-by-point responses to reviewers' comments. If you are prepared to undertake the work required, I would be pleased to reconsider my decision. Please note that this revision decision does not assure the acceptance of your work. Thanks for the opportunity to consider your work.

Reviewers' comments:

Reviewer's Responses to Questions

**Comments to the Author**

1. Is the manuscript technically sound, and do the data support the conclusions?

Reviewer #1: Yes

Reviewer #2: Partly

Reviewer #3: Partly

Reviewer #4: Yes

Reviewer #5: Yes

2. Has the statistical analysis been performed appropriately and rigorously?

Reviewer #1: Yes

Reviewer #2: No

Reviewer #3: I Don't Know

Reviewer #4: No

Reviewer #5: Yes

3. Have the authors made all data underlying the findings in their manuscript fully available?

Reviewer #1: Yes

Reviewer #2: Yes

Reviewer #3: Yes

Reviewer #4: Yes

Reviewer #5: Yes

4. Is the manuscript presented in an intelligible fashion and written in standard English?

Reviewer #1: Yes

Reviewer #2: Yes

Reviewer #3: Yes

Reviewer #4: Yes

Reviewer #5: Yes

Reviewer #1: The manuscript presents a well-structured computational framework that integrates single-cell RNA-seq deconvolution with weakly-supervised, attention-based multiple-instance learning to infer immune-cell composition directly from H&E whole-slide images (WSIs) of hepatocellular carcinoma. The work is timely, the methodology is technically sound, and the results demonstrate a clear improvement over the CLAM baseline. Nevertheless, several substantive issues—ranging from terminology precision to external validation—must be addressed before the paper can be considered for publication in PLOS ONE. I recommend Major Revision.

Major Comments

1. Throughout the manuscript the authors refer to “liver cancer” or “LIHC” without specifying that the study is restricted to HCC. Because intrahepatic cholangiocarcinoma and combined HCC-ICC have distinct immune microenvironments.

2. The Abstract and Discussion repeatedly label the approach as “non-invasive.” Since WSIs are derived from surgically resected or biopsied specimens, the technique is technically post-operative/ex vivo, not non-invasive.

3. The study relies solely on publicly available TCGA-LIHC WSIs and GEO scRNA-seq data. The manuscript currently lacks independent clinical validation (e.g., a local HCC cohort with flow-cytometry or multiplex-IHC ground truth). Please add prospective validation with HCC specimens.

4. Figure 5A (UMAP after log-normalization) is supplied at insufficient resolution. Please provide a high-resolution vector graphic.

5. Report exact p-values (not “p < 0.05”) for all log-rank tests in Supplementary Figure 10E–H and include 95 % confidence intervals for the accuracy metrics in Table 1.

6. Provide a brief justification for selecting log-normalization over CCA or SCTransform, despite the latter showing tighter batch correction.

Reviewer #2: The manuscript presents a timely and technically innovative framework (LIMPACAT) that integrates single-cell RNA-seq data and whole-slide imaging (WSI) to predict immune cell compositions in liver cancer. The multi-omics attention transformer approach is novel and shows promising results, particularly in stratifying patients by immune cell infiltration with implications for prognosis.

However, several important concerns should be addressed before the manuscript can be considered for publication:

1- Technical validity and data support:

The manuscript is partially supported by the data. While the use of deconvolution from scRNA-seq to simulate bulk RNA-seq is methodologically reasonable, there is no direct experimental validation of the immune cell predictions (e.g., via flow cytometry, IHC, or spatial transcriptomics). Moreover, all model training and evaluation were conducted using a single dataset (TCGA-LIHC), limiting generalizability.

2 -Statistical analysis:

Statistical approaches are broadly appropriate but lack sufficient rigor and transparency. There is no report of confidence intervals or statistical significance for model accuracy, nor of hazard ratios in survival analyses. No corrections for multiple hypothesis testing are mentioned, and data splitting or cross-validation strategies for model training are not described, raising concerns about potential overfitting.

3- Language and presentation:

The manuscript is not yet written in fully standard English. While the technical content is generally understandable, there are multiple grammatical and syntactic issues that require revision. Additionally, some sections (especially Methods and Discussion) are repetitive or overly verbose. A professional language edit is strongly recommended to improve clarity and readability.

In summary, this manuscript presents a creative and impactful contribution to digital pathology and computational immuno-oncology. With improved statistical rigor, clearer methodological reporting, and stronger language polishing, it has the potential to make a meaningful contribution to the field.

Reviewer #3: The authors present LIMPACAT (Liver Immune Microenvironment Prediction and Classification Attention Transformer), a framework that utilizes whole-slide images to predict immune cell compositions relevant to liver cancer prognosis. The concept is innovative; however, the manuscript is very difficult to follow in many key areas and it requires significant revision before it can be adequately evaluated.

Major Comments:

1. Clarity of Model Architecture and Workflow:

The inputs and outputs of the MIL-ATTENTION model are unclear in Figure 1 and throughout the manuscript. What is the output of the MIL-ATTENTION model? Figure 1 should also clearly distinguish between the training pipeline and the inference pipeline for both the ensemble DL-CCD model and the MIL-ATTENTION model, including the type of data and processing applied in each.

2. Positioning Against Existing Methods:

Many deconvolution models exist for bulk RNA-seq. The authors need to clearly articulate the advantages of their DL-CCD model over established methods.

3. Reorganization of the Results Section:

The sections from “Gene Count and Quality Control Metrics” to “Single-Cell Type Annotation” are too detailed for the Results section and should be condensed into a brief summary, with the full methodological descriptions moved to the Methods section. The Results should highlight findings that are novel and relevant to the field.

4. Validation Using Real Bulk RNA-seq Data:

The evaluation of the cell deconvolution model appears to rely primarily on simulated bulk RNA-seq derived from scRNA-seq. However, scRNA-seq and actual bulk RNA-seq have distinct characteristics. The model should be validated on real-world bulk RNA-seq datasets with known or estimated cell compositions to establish robustness.

5. Clarity and Framework Usability:

The section titled “Cell Composition Deconvolution Model and Immune Cell Level Estimated from WSI” is currently difficult to follow and lacks detail. It is the core of the paper and should be rewritten to clearly describe how the model components interact. It should also contain information about how the overall framework can be used by others.

6. Overfitting and Generalization:

There appears to be overfitting across all cell types. The authors should address this issue and describe any strategies used to mitigate overfitting (e.g., regularization, validation, dropout, etc.).

Minor Comments:

7. Acronyms such as “WSIs” and “CCD” should be spelled out at first mention. This is especially important in the abstract and background sections.

8. Several sentences in the Background section (e.g. paragraphs 2 and 3) lack citations.

9. Why using CCA and SCT normalization on LIHC bulk RNA-seq samples?

Reviewer #4: I read the submission titled “LIMPACAT:Multi-Omics Attention Transformer for Immune Prediction in Liver Cancer Using Whole-Slide Imaging” and thank the author for this interesting piece of original research. This manuscript presents a novel and technically compelling framework that integrates single-cell RNA-seq, simulated bulk transcriptomics, and transformer-based MIL models to predict immune composition from WSIs in HCC. The methodology is creative, well-motivated, and appears technically sound. Several important checks, such as the benchmarking of normalization methods and MIL model comparisons are included.

However, I believe the structure of the paper can be improved and additional information is necessary to make results reproducible and increase accessibility to a wider audience:

1. The methodology section is short and several important methodological steps are only introduced in the results. The first paragraphs of the results section do not contain any results and instead focus on quality control steps, software libraries and clustering methods. A clearer separation of methods and results would improve readability.

2. The data sets used are insufficiently characterized. I could not find information on the survival analysis data set, including number of patients that had matched WSIs and survival data available, their demographic and disease characteristics, the duration of follow up and whether there was any clustering present that could influence the statistical analysis (is there one slide per patient or potentially multiple?).

3. It is not entirely clear why only 20 samples were chosen from the GSE189903 dataset. This is generally a very small sample. Were the other 14 from different types of liver cancer (that would be a very high proportion) or was this number based on statistical or computational considerations? Do the 20 samples correspond to 20 different patients?

4. The validation using survival analysis is insufficiently described and could be more thorough. It is not clear how low and high CCD prediction groups were determined, where the cutoff was set and why CCD was not validated as a continuous predictor. Likewise, the paper mentions good and poor survival groups, but does not describe how they were determined. Using the raw prediction scores and survival times directly for validation would have been better. Instead of accuracy the C-index could have been reported. The method for splitting the sample into train/test or cross validation folds is not described. Uncertainty should be quantified. Alternative explanations for the reported associations should at least be discussed. A multivariate model adjusted for (at least) tumor stage could improve interpretability.

Reviewer #5: This study presents LIMPACAT, a novel and timely deep learning framework for predicting clinically relevant immune cell populations in liver cancer directly from whole-slide images. The approach of integrating single-cell and bulk transcriptomics with digital pathology is at the forefront of computational oncology. The findings that higher B and NK cell levels correlate with better prognosis are consistent with existing literature and demonstrate the potential of this tool for non-invasive biomarker discovery. While the work is promising, the following points should be addressed to strengthen the manuscript.

Major Revisions

1.Validation of Deconvolution Predictions: The most significant limitation of this study is the reliance on computationally inferred immune cell compositions as ground truth labels without subsequent biological validation. The model's high accuracy indicates it can successfully predict these inferred labels, but whether these labels accurately reflect the true cellular makeup of the TME is unknown. The authors should:

① More explicitly and prominently state this limitation in the Discussion and Conclusion sections.

② Suggest future work involving validation against a ground truth, such as performing immunohistochemistry (IHC) or multiplex immunofluorescence (mIF) for the predicted cell types (B cells, NK cells) on a subset of the WSI slides to correlate cell counts. For an example of how such validation is performed, the authors could refer to recent studies that correlate computational predictions with IHC.

2. Comparison with State-of-the-Art: The discussion compares LIMPACAT to several other models, including CLAM. However, the field of transformers in computational pathology is advancing rapidly. The manuscript would be significantly strengthened by discussing and contextualizing LIMPACAT with other very recent (2023-2025) transformer-based architectures for WSI analysis and immune prediction. This would provide a clearer picture of where LIMPACAT stands in the current landscape.

Minor Revisions

1. Improvement of figure quality:

The images of Supplementary Figure 10 (A)-(E) were exported out rather low resolution. Please ensure image clarity.

2.Methodological Clarity:

①In the Methods section, please specify the exact threshold or method used to stratify patients into "high" and "low" immune cell groups for the survival analysis (e.g., median, quartiles).

②Provide a more detailed description of the custom CNN, ATT, and ATT_TRANS model architectures, either in the main text or as a supplementary methods section.

**Do you want your identity to be public for this peer review?** For information about this choice, including consent withdrawal, please see our Privacy Policy

Reviewer #1: **Yes: ** Yigang Zhang

Reviewer #2: **Yes: ** VALERIA DUARTE DE ALMEIDA

Reviewer #3: No

Reviewer #4: No

Reviewer #5: No

---

## [Author Response · Author response to Decision Letter 1]

20 Oct 2025

Journal Requirements:

and

Response 1

We thank the editor for pointing this out. We have checked and revised the formatting. Author name and affiliation formatting will be adjusted according to the PLOS ONE style. Supplementary figure labels and in-text citations have been updated for consistency with the manuscript.

Response 2

Software and data availability

We thank the editor for this reminder. The Software and Data Availability section has been revised. All custom code is openly available at GitHub (https://github.com/holiday01/LIMPACAT.git) and archived in Zenodo (10.5281/zenodo.17168482).

This work was supported by the grants from the National Science and Technology Council (NSTC).

This research was supported by grants from the National Science and Technology Council (NSTC), Taiwan (NSTC 112-2222-E-130-003- and NSTC 113-2221-E-130-005-MY3), which assisted in the acquisition of computational facilities essential for this study.

Response 3

We thank the editor for highlighting this. The Funding section has been revised to include full grant information.

No available

Response 4

We thank the editor for the clarification. The Competing Interests section has been revised to state that the authors declare no competing interests.

5. In the online submission form, you indicated that your data will be submitted to a repository upon acceptance. We strongly recommend all authors deposit their data before acceptance, as the process can be lengthy and hold up publication timelines. Please note that, though access restrictions are acceptable now, your entire minimal dataset will need to be made freely accessible if your manuscript is accepted for publication. This policy applies to all data except where public deposition would breach compliance with the protocol approved by your research ethics board. If you are unable to adhere to our open data policy, please kindly revise your statement to explain your reasoning and we will seek the editor's input on an exemption.

Response 5

We thank the editor for this note. All datasets generated in this study have been deposited and are openly accessible at GitHub (https://github.com/holiday01/LIMPACAT.git) and archived in Zenodo (10.5281/zenodo.17168482). These include derived datasets such as TCGA-LIHC WSI file identifiers, train/validation/test splits, and deconvolution labels. For external datasets, the raw WSIs are publicly available from the GDC Data Portal (https://portal.gdc.cancer.gov/), and the RNA-seq dataset is available under GEO accession, with download instructions provided by the respective repositories.

6. Please amend your list of authors on the manuscript to ensure that each author is linked to an affiliation. Authors’ affiliations should reflect the institution where the work was done (if authors moved subsequently, you can also list the new affiliation stating “current affiliation:….” as necessary).

Response 6

We thank the editor for the reminder. All author affiliations have been carefully checked and revised to ensure that each author is explicitly linked to an institution. Current affiliations have also been indicated where applicable.

7. Please upload a new copy of Figure 1 as the detail is not clear. Please follow the link for more information: https://blogs.plos.org/plos/2019/06/looking-good-tips-for-creating-your-plos-figures-graphics/"" https://blogs.plos.org/plos/2019/06/looking-good-tips-for-creating-your-plos-figures-graphics/

Response 7

We thank the editor for this important reminder. All supplementary figures and tables have been consolidated into a single file. Each element is labeled according to PLOS ONE style (e.g., S1 Fig, S2 Fig, S1 Table). In-text citations have been revised accordingly.

Response 8

We thank the editor for this important reminder. All supplementary figures and tables have been consolidated into a single file (“S1 File. Supplemental Figures and Tables”). Each element is labeled according to PLOS ONE style (e.g., S1 Fig, S2 Fig, S1 Table). In-text citations have been revised accordingly.

Response 9

We thank the editor for this note. We have reviewed the suggested references. Relevant publications will be cited where appropriate, while others will be carefully evaluated in the context of our study.

Reviewer #1

1. Throughout the manuscript the authors refer to “liver cancer” or “LIHC” without specifying that the study is restricted to HCC. Because intrahepatic cholangiocarcinoma and combined HCC-ICC have distinct immune microenvironments.

Reviewer #1Response1

We appreciate the reviewer for this valuable comment. We have revised the manuscript to clarify that all analyses are restricted to hepatocellular carcinoma (HCC). Ambiguous terms such as “liver cancer” have been replaced with “HCC” throughout the revised manuscript. We explicitly indicate that the GSE189903 dataset comprises 20 HCC samples and clarify that the TCGA-LIHC cohort, as defined by The Cancer Genome Atlas, includes only HCC cases, excluding intrahepatic cholangiocarcinoma (ICC) and combined HCC-ICC cases. (Line 136-139)

2. The Abstract and Discussion repeatedly label the approach as “non-invasive.” Since WSIs are derived from surgically resected or biopsied specimens, the technique is technically post-operative/ex vivo, not non-invasive.

Reviewer #1Response 2

We appreciate the reviewer for this important clarification. We agree that whole-slide images (WSIs) are derived from surgically resected or biopsied tissues and, therefore, the approach should not be described as “non-invasive.” Consequently, we have revised the Abstract and Discussion to more precisely state that our method enables ex vivo immune profiling and pathology-based tumor microenvironment characterization. (Line 12)

3. The study relies solely on publicly available TCGA-LIHC WSIs and GEO scRNA-seq data. The manuscript currently lacks independent clinical validation (e.g., a local HCC cohort with flow-cytometry or multiplex-IHC ground truth). Please add prospective validation with HCC specimens.

Reviewer #1 Response 3

We appreciate the reviewer for highlighting this important point. We acknowledge that the lack of prospective validation using a local HCC cohort with flow cytometry or multiplex immunohistochemistry (mIHC) reference data constitutes a limitation of this study. To address this concern, we implemented a two-step validation strategy. First, we used HCC single-cell RNA sequencing (scRNA-seq) data to generate simulated bulk RNA-seq samples, demonstrating the feasibility of the cell composition deconvolution (CCD) framework. (Line 403-424)

Second, in this revision, we added to perform independent validation using newly generated peripheral blood mononuclear cell (PBMC) bulk RNA-seq data with experimentally defined immune cell populations. This additional dataset confirmed the robustness of the CCD model beyond simulated data (Line 422-424).

4. Figure 5A (UMAP after log-normalization) is supplied at insufficient resolution. Please provide a high-resolution vector graphic.

Reviewer #1Response 4

We thank the reviewer for the comment. Figure 5A (UMAP after log-normalization) has been revised and is now provided as a high-resolution vector graphic.

5. Report exact p-values (not “p < 0.05”) for all log-rank tests in Supplementary Figure 10E–H and include 95 % confidence intervals for the accuracy metrics in Table 1.

Reviewer #1Response 5

We appreciate the reviewer for this insightful suggestion. Ninety-five percent confidence intervals for all accuracy metrics in Table 1 have been included, calculated from the sample mean and standard error based on 30 repeated model training runs using train-validation splits. Additionally, the exact p-values for all log-rank tests have now been reported. Owing to resolution issues, the corresponding plots have been reformatted and are now presented in S11 Fig.

6. Provide a brief justification for selecting log-normalization over CCA or SCTransform, despite the latter showing tighter batch correction.

Reviewer #1Response 6

Thank you for your valuable comment. As outlined in the Results section under “Cell Composition Deconvolution,” we compared three normalization methods: CCA, SCT, and log normalization. Log normalization demonstrated the highest clustering consistency, as measured using the adjusted Rand index (ARI), and preserved greater biological variability across samples, which is essential for capturing tumor microenvironment heterogeneity. These findings, consistent with those of ARI metrics and correlation analyses (S10 Fig), informed our decision to use the log-normalized CCD model for downstream survival analyses. (Line 343-355)

Reviewer #2

1- Technical validity and data support:

The manuscript is partially supported by the data. While the use of deconvolution from scRNA-seq to simulate bulk RNA-seq is methodologically reasonable, there is no direct experimental validation of the immune cell predictions (e.g., via flow cytometry, IHC, or spatial transcriptomics). Moreover, all model training and evaluation were conducted using a single dataset (TCGA-LIHC), limiting generalizability.

Reviewer #2 Response 1

We acknowledge that our study does not include direct experimental validation, such as flow cytometry, IHC, or spatial transcriptomics, which would provide the most definitive confirmation of immune cell proportions.

However, beyond using simulated bulk RNA-seq generated from scRNA-seq—a widely adopted proxy supported by previous studies demonstrating reasonable concordance with experimental measurements—we conducted external validation using real-world PBMC bulk RNA-seq data from the GSE107011 dataset. This dataset comprises bulk RNA-seq profiles of sorted immune cell populations with well-characterized compositions, enabling us to evaluate model performance against experimentally defined ground truth. The comparison demonstrated strong concordance, quantified using Pearson correlation coefficients (PCCs), thereby supporting the robustness of our deconvolution model in real bulk RNA-seq contexts. (Line 251-255, 419-427)

We have revised the Discussion to highlight both the absence of direct experimental validation in liver cancer tissues and the inclusion of PBMC-based validation, partially mitigating this limitation. (Line 526-540)

2 -Statistical analysis:

Statistical approaches are broadly appropriate but lack sufficient rigor and transparency. There is no report of confidence intervals or statistical significance for model accuracy, nor of hazard ratios in survival analyses. No corrections for multiple hypothesis testing are mentioned, and data splitting or cross-validation strategies for model training are not described, raising concerns about potential overfitting.

Reviewer#2Response 2

We appreciate the reviewer for this valuable suggestion. To enhance the statistical transparency of our study, we have clarified the evaluation metrics used for each analysis. For the deconvolution model, performance was assessed using the PCC, a standard commonly used to benchmark cell composition predictions. Additionally, we present results derived from both simulated and real bulk RNA-seq datasets (GSE107011). (Line 251-255, 419-427)

For the WSI-based classification analyses, we have added 95% confidence intervals to all accuracy metrics in Table 1, calculated from the sample mean and standard error across 30 repeated training runs with train–validation splits. For the survival analyses, the exact p-values of all log-rank tests are now reported. Corresponding plots have been reformatted and moved to S11 Fig. for improved clarity. These revisions enhance the rigor and transparency of our statistical reporting. (Line 262-266)

3- Language and presentation:

The manuscript is not yet written in fully standard English. While the technical content is generally understandable, there are multiple grammatical and syntactic issues that require revision. Additionally, some sections (especially Methods and Discussion) are repetitive or overly verbose. A professional language edit is strongly recommended to improve clarity and readability.

In summary, this manuscript presents a creative and impactful contribution to digital pathology and computational immuno-oncology. With improved statistical rigor, clearer methodological reporting, and stronger language polishing, it has the potential to make a meaningful contribution to the field.

Reviewer#2Response 3

We appreciate the reviewer’s comment regarding the language quality. The manuscript has already undergone professional English editing, and the revised version with proof of editing has been uploaded with this submission.

Reviewer #3

Major Comments:

1. Clarity of Model Architecture and Workflow:

The inputs and outputs of the MIL-ATTENTION model are unclear in Figure 1 and throughout the manuscript. What is the output of the MIL-ATTENTION model? Figure 1 should also clearly distinguish between the training pipeline and the inference pipeline for both the ensemble DL-CCD model and the MIL-ATTENTION model, including the type of data and processing applied in each.

Reviewer #3Response1

We appreciate the reviewer for this valuable comment. In response, we have revised both the text and Fig 1 to more clearly illustrate the architecture and workflow of the MIL-ATTENTION model and its integration with the ensemble DL-CCD framework. Specifically, we have updated Figure 1 to distinctly separate the training pipeline (including scRNA-seq–derived pseudo-bulk RNA-seq for deconvolution, generation of immu

---

## [Decision Letter · Decision Letter 1]

26 Nov 2025

Dear Dr. Chiu,

Thank you for submitting your manuscript to PLOS ONE. After careful consideration, we feel that it has merit but does not fully meet PLOS ONE’s publication criteria as it currently stands. Therefore, we invite you to submit a revised version of the manuscript that addresses the points raised during the review process.

We look forward to receiving your revised manuscript.

Kind regards,

Li Yang, M.D.

Academic Editor

PLOS ONE

Journal Requirements:

Additional Editor Comments:

Please further address the reviewer's minor concerns.

Reviewers' comments:

Reviewer's Responses to Questions

**Comments to the Author**

Reviewer #1: All comments have been addressed

Reviewer #3: (No Response)

Reviewer #4: All comments have been addressed

2. Is the manuscript technically sound, and do the data support the conclusions?

Reviewer #1: Yes

Reviewer #3: Partly

Reviewer #4: Yes

3. Has the statistical analysis been performed appropriately and rigorously?

Reviewer #1: Yes

Reviewer #3: I Don't Know

Reviewer #4: Yes

4. Have the authors made all data underlying the findings in their manuscript fully available?

Reviewer #1: Yes

Reviewer #3: Yes

Reviewer #4: Yes

5. Is the manuscript presented in an intelligible fashion and written in standard English?

Reviewer #1: Yes

Reviewer #3: Yes

Reviewer #4: Yes

Reviewer #1: The authors have addressed all of my concerns in a detailed and rigorous manner. They have made the necessary revisions to the manuscript, and I believe that the current version of the article is now acceptable for publication.

Reviewer #3: Many places in the manuscript contain "Error! Reference source not found" that should be fixed.

Re Reviewer #3Response2, please point out the paragraph that "explicitly compare DL-CCD with established bulk RNA-seq deconvolution methods, such as quanTIseq and CIBERSORTx".

Reviewer #4: My comments were adequately addressed. I support acceptance given the additional inclusion of a separate validation cohort.

**Do you want your identity to be public for this peer review?** For information about this choice, including consent withdrawal, please see our Privacy Policy

Reviewer #1: **Yes: ** Yigang Zhang

Reviewer #3: No

Reviewer #4: No

---

## [Author Response · Author response to Decision Letter 2]

27 Nov 2025

Reviewer #1: The authors have addressed all of my concerns in a detailed and rigorous manner. They have made the necessary revisions to the manuscript, and I believe that the current version of the article is now acceptable for publication.

Response Reviewer #1:

We thank the reviewer for the positive evaluation and for acknowledging the improvements made in the revised manuscript.

Reviewer #3: Many places in the manuscript contain "Error! Reference source not found" that should be fixed.

Response Reviewer #3:

We thank the reviewer for identifying this issue. All cross-reference and citation errors have been carefully reviewed and fully corrected throughout the revised manuscript. No remaining “Error! Reference source not found” messages appear in the updated version.

Re Reviewer #3Response2, please point out the paragraph that "explicitly compare DL-CCD with established bulk RNA-seq deconvolution methods, such as quanTIseq and CIBERSORTx".

Response Re Reviewer #3Response2:

We thank the reviewer for the comment. The previous discussion referred to comparisons with conventional methods such as quanTIseq and CIBERSORTx through cited studies, but this was not explicitly stated. We have now added a clear sentence in the Introduction to explicitly indicate these methods.

The following text has been added in the Introduction (Lines 51–58):

“Early CCD models relied on regression model frameworks such as quanTIseq and CIBERSORTx, which depend on predefined immune signatures and therefore struggle to capture nonlinear gene–cell relationships and tumor-specific immune heterogeneity. More recent studies have demonstrated that incorporating single-cell RNA sequencing (scRNA-seq) references with deep learning (DL) architectures substantially improves deconvolution accuracy. In our previous work, we introduced a DL-based CCD model that achieved higher concordance with true immune cell fractions than both CIBERSORTx and quanTIseq [8].”

Reviewer #4: My comments were adequately addressed. I support acceptance given the additional inclusion of a separate validation cohort.

Response Reviewer #4:

We sincerely thank the reviewer for the positive feedback and for recognizing the improvements made in the revision. We appreciate the reviewer’s earlier comments, which helped strengthen our validation strategy and clarify the contribution of our method.

---

## [Decision Letter · Decision Letter 2]

10 Dec 2025

LIMPACAT:Multi-Omics Attention Transformer for Immune Prediction in Liver Cancer Using Whole-Slide Imaging

PONE-D-25-22455R2

Dear Dr. Chiu,

We’re pleased to inform you that your manuscript has been judged scientifically suitable for publication and will be formally accepted for publication once it meets all outstanding technical requirements.

Kind regards,

Li Yang, M.D.

Academic Editor

PLOS One

Additional Editor Comments (optional):

Thanks for the authors' efforts to comprehensively improve your manuscript according to editor's and reviewers' comments. I am pleased to inform you that your paper can be accepted for publication now.

Reviewers' comments:

Reviewer's Responses to Questions

**Comments to the Author**

Reviewer #3: All comments have been addressed

2. Is the manuscript technically sound, and do the data support the conclusions?

Reviewer #3: Yes

3. Has the statistical analysis been performed appropriately and rigorously?

Reviewer #3: Yes

4. Have the authors made all data underlying the findings in their manuscript fully available?

Reviewer #3: Yes

5. Is the manuscript presented in an intelligible fashion and written in standard English?

Reviewer #3: Yes

Reviewer #3: My comments were adequately addressed. The current version of the article is acceptable for publication.

**Do you want your identity to be public for this peer review?** For information about this choice, including consent withdrawal, please see our Privacy Policy

Reviewer #3: No

---

## [Editor Report · Acceptance letter]

PONE-D-25-22455R2

PLOS One

Dear Dr. Chiu,

I'm pleased to inform you that your manuscript has been deemed suitable for publication in PLOS One. Congratulations! Your manuscript is now being handed over to our production team.

Kind regards,

on behalf of

Dr. Li Yang

Academic Editor

PLOS One